# Rural–Urban Inequalities in Poor Self-Rated Health, Self-Reported Functional Disabilities, and Depression among Chinese Older Adults: Evidence from the China Health and Retirement Longitudinal Study 2011 and 2015

**DOI:** 10.3390/ijerph18126557

**Published:** 2021-06-18

**Authors:** Haiting Jiang, Bo Burström, Jiaying Chen, Kristina Burström

**Affiliations:** 1School of Health Policy and Management, Nanjing Medical University, No.101 Longmian Avenue, Nanjing 211166, China; jianghaiting93@126.com; 2Centre for Health Policy Studies, Nanjing Medical University, No.101 Longmian Avenue, Nanjing 211166, China; bo.burstrom@ki.se (B.B.); kristina.burstrom@ki.se (K.B.); 3Health Outcomes and Economic Evaluation Research Group, Stockholm Centre for Healthcare Ethics, Department of Learning, Informatics, Management and Ethics, Karolinska Institutet, 17177 Stockholm, Sweden; 4Equity and Health Policy Research Group, Department of Global Public Health, Karolinska Institutet, 17177 Stockholm, Sweden; 5Institute of Healthy Jiangsu Development, No.101 Longmian Avenue, Nanjing 211166, China

**Keywords:** China, functional ability, health inequalities, Hukou, older adults, rural–urban, self-reported health, social determinants of health

## Abstract

The household registration system (Hukou) in China classifies persons into rural or urban citizens and determines eligibility for state-provided services and welfare. Not taking actual residence into account may underestimate rural–urban differences. This study investigates rural–urban inequalities in self-reported health outcomes among older adults aged 60+, taking into account both Hukou and actual residence, adjusting for sociodemographic determinants, based on the China Health and Retirement Longitudinal Study (CHARLS) in 2011 and 2015. Self-Rated Health (SRH) was assessed with a single question, functional abilities were assessed with the Basic Activities of Daily Living (BADLs) and Instrumental Activities of Daily Living (IADLs) scales, and depression was assessed with the 10-item version of the Center for Epidemiologic Studies Depression Scale. Rural respondents had poorer socioeconomic status and higher prevalence of poor SRH, functional disabilities, and depression than urban respondents in both years, which were closely related to rural–urban differences in educational level and income. Impairments appeared at a younger age among rural respondents. Analyses using only Hukou registration and not actual residence resulted in underestimation of rural–urban differences. This study may serve as a basis for interventions to address rural–urban differences in health and social services and reduce health inequalities among Chinese older adults.

## 1. Introduction

The proportion of older adults (60 years and above) in China is increasing [1,2], comprising 18.1% of the total population in 2019. The aging of the population will increase the need and demand for health and social services. Nearly 60% of Chinese older adults live in rural areas [3,4]. Rural areas are less economically developed and older adults remain at home while their children move to urban areas for work [3]. Compared to older adults in urban areas, those in rural areas have lower socioeconomic status [4], less access to health services and social support [3], and report worse health status [5]. The household registration system (Hukou) in China, which classifies each person into a rural or an urban citizen, is a major means of monitoring population mobility and determining eligibility for state-provided welfare and services [6]. Citizens’ interests and rights, such as the right to education, health insurance, pension insurance, housing and employment, welfare and social security are determined by their Hukou registration [7]. Persons with rural Hukou have poorer entitlements in all these aspects than those with urban Hukou, which together with other disparities in living conditions may contribute to rural–urban inequalities in health outcomes [8,9,10]. In addition, a person with rural Hukou registration may live in an urban area and vice versa, and there is considerable in-migration to urban areas from rural areas [9,11]. Hence, when comparing rural and urban residents, using only the Hukou registration and not actual residence may result in misclassification. Those with rural Hukou migrating to urban areas tend, for instance, to have better health than those remaining in rural areas [11], but this is likely explained by persons migrating being younger and healthier persons. Some previous studies have taken Hukou and population mobility into consideration [9,10,11]. However, few studies have focused on rural–urban differentials in the health of older adults, taking into account both Hukou and residence classification.

The aging of the population places increased demands on the provision of health and social services; which are less developed in rural areas., However, because of a greater accumulation of adverse exposures and events over the life course, the need for such services may arise at lower age in rural areas [10]. The Hukou registration system is currently under discussion, and changes may be underway to improve entitlements for those with rural Hukou [12]. An expansion of health and social services will require more knowledge regarding which needs older people have, especially in rural areas. It may also be useful for planners to know the trends over time of the needs to be addressed. Different dimensions of health, such as functional abilities [13] and psychological health [14], are important for an individual’s well-being [15]. Functional abilities deteriorate as people become older [16]. Living alone or not [17,18,19] and the proximity to children could influence the older adults’ health [18,20]. Furthermore, socioeconomic factors are important determinants of health associated to differences in health between rural and urban older adults [5,21,22], suggesting that wider policy interventions may be needed to address the social determinants of health so as to reduce rural–urban inequalities in health.

The present study investigated rural–urban inequalities in 2011 and 2015 in the prevalence of poor self-rated health, self-reported functional disabilities, and depression among Chinese older adults, taking into account both Hukou and actual residence classification, and analyzed sociodemographic determinants of rural–urban inequalities in those self-reported health outcomes.

## 2. Materials and Methods

### 2.1. Data Sources

The China Health and Retirement Longitudinal Study (CHARLS) is a nationally representative household survey of the Chinese population, conducted by Peking University [23]. The baseline survey was conducted between June 2011 and March 2012 covering 28 provinces and 17,708 respondents (age ≥45 years) from 10,257 households. Two follow-up interviews were conducted in 2013 and 2015. Data are publicly available. Our study used cross-sectional data of CHARLS 2011 and CHARLS 2015 to investigate the prevalence of poor self-rated health (SRH), impaired Basic Activities of Daily Living (BADLs), and impaired Instrumental Activities of Daily Living (IADLs) and self-reported depression among rural and urban Chinese older adults aged 60 years and above. There were 7638 respondents in CHARLS 2011 and 10,185 respondents in CHARLS 2015 who were included in our study.

### 2.2. Demographic Indicators

In order to avoid misclassification of individuals in rural and urban areas, we combined the information on Hukou registration with information on actual residence. Respondents who have rural Hukou registration (agricultural household) and live in rural areas were identified as rural respondents. Respondents who have urban Hukou registration (non-agricultural household) and live in urban areas were considered as urban respondents. Two additional groups were identified with discordant Hukou registration and residence. Age was divided into age groups: 60–64, 65–69, 70–74, 75–79, 80–84, 85–89, and 90+ years. Marital status was dichotomized into married and unmarried (i.e., widowed, never married, divorced, or separated). Living arrangement was categorized into living alone or not living alone. In CHARLS, respondents were asked whether they had a child who lives in the same city or county. Respondents co-residing with a child, or whose child lived in the same city or county were defined as “living near children”. Respondents with children not co-residing or living in the same city or county were defined as “not living near children”. Respondents with no children were categorized as “no child”.

### 2.3. Socioeconomic Indicators

Educational level was based on highest self-reported attained education and categorized into below primary school, primary school, middle school, high school, and college and above. An individual’s annual income was assessed by dividing the total household annual income by the number of persons living in the family within the last half-year, regardless of age and employment status [24]. Total annual household income is the sum of all income from all household members, including income from earnings, capital income, pension, government transfers, and other income. Respondents were then ranked from lowest to highest by their annual income and divided into five groups of equal size. In CHARLS 2011, the lowest income group had an income below CNY 610 (Chinese Yuan); the second group from CNY 611 to 2100; the third group from CNY 2101 to 5325; the fourth group from CNY 5326 to 12,067; the fifth and highest income group CNY 12,068 and above. In CHARLS 2015, the lowest income group had an income below CNY 500; the second group from CNY 501 to 1169; the third group from CNY 1170 to 3599; the fourth group from CNY 3600 to 13,339; the fifth and highest income group CNY 13,440 and above.

### 2.4. Outcome Variables

#### 2.4.1. Self-Rated Health

Self-rated health (SRH) is a generic measurement of health [25,26]. SRH was measured by a single question. CHARLS adopted two different 5-point scales for self-reported general health, which were randomly assigned to participants to examine any effects of central tendency bias, i.e., that respondents rated their health on either the scale “excellent”, “very good”, “good”, “fair” and “poor” or on the scale “very good”, “good”, “fair”, “poor” and “very poor”. Respondents were asked twice about their health status, once in the beginning of the health module with one scale and again at the end of the health module with the other scale. This study focused on persons with poor health outcomes, so the scale ranging from “very good” to “very poor” was used. Those who answered poor or very poor health were categorized as having poor health.

#### 2.4.2. Basic Activities of Daily Living

Katz Activities of Daily Living Scale [27] was used to assess Basic Activities of Daily Living (BADLs). It is a 6-item scale with dressing, bathing and showering, eating, getting in and out of bed, using the toilet, and controlling urination and defecation. Respondents were asked “Do you have any difficulty with the following basic activity of daily living?” With the scoring system used in CHARLS, each item was scored as follows: 1 signified “do not have any difficulty”, 2 signified “have difficulties but still can do it”, 3 signified “have difficulties and help is needed”, and 4 signified “cannot complete it”. The Chinese version of the scale has been extensively tested and shown to yield reliable and valid responses [28]. The score of BADLs was calculated by the sum of all items. Having any difficulty with an activity (total score of BADLs >6) was identified as “impaired BADLs”.

#### 2.4.3. Instrumental Activities of Daily Living

Instrumental Activities of Daily Living (IADLs) were measured by the Lawton IADL Scale [29], which is ideal for community-dwelling older adults [30]. Performance was examined on the 5-items scale with doing household chores, cooking, shopping, managing money, taking medications [29]. Respondents were asked “Do you have any difficulty with the following instrumental activity of daily living?” With the scoring system used in CHARLS, each item was scored as following: 1 signified “do not have any difficulty”, 2 signified “have difficulties but still can do it”, 3 signified “have difficulties and help is needed”, and 4 signified “cannot complete it”. A sum score of all items >5 was categorized as “impaired IADLs”.

#### 2.4.4. Self-Reported Depression

In CHARLS, the 10-item version of the Center for Epidemiologic Studies Depression Scale (CES-D-10) was used to measure respondents’ self-reported depression. CES-D-10 score is the sum of the 10 self-reported questions, after reverse coding. Additionally, the scale for each of the 10 questions was adjusted so that the response options were 0 to 3, CES-D-10 ranges from 0 to 30 with higher scores indicating that the respondent felt more negatively during the past week. Participants were categorized as having depressive symptoms if their total score on the CES-D-10 was 10 or above [31].

### 2.5. Statistical Analyses

Respondents with missing data on Hukou registration, sex, marital status, living arrangement, and educational level were excluded. In an initial step, analyses were performed on the prevalence rate of the health outcomes studied among respondents having different Hukou registration in rural and urban areas.

The main analyses in this study focused on rural respondents versus urban respondents. Hence, we excluded in the main analyses the respondents with rural Hukou registration living in urban areas and respondents with urban Hukou registration living in rural areas. Therefore, the final samples used for analyses were 6048 respondents in CHARLS 2011 and 7396 respondents in CHARLS 2015 (Figure 1).

Descriptive statistics are presented as means with standard deviation (SD) for numerical variables, or as percentages for categorical variables. Differences in the prevalence of poor SRH, impaired BADLs, impaired IADLs, and self-reported depression between the rural and urban respondents were examined separately for 2011 and 2015, by using Chi-square test or Fisher’s Exact test for nominal categorical variables [32] and for variables with ordinal explanatory variables. Independent *t* test was used to analyze the differences between rural and urban respondents in means of age and income. Multiple logistic regression models were performed to examine the odds ratio (OR) of poor SRH, functional disabilities, and depression among rural respondents and to what extent this over-risk was explained by sociodemographic determinants, adjusting for age and sex.

All statistical analyses were performed using SAS 9.4.1 (SAS Institute Inc., Cary, NC, USA). The level of significance was specified at 0.05. A Bonferroni adjusted significance level was used in case of multiple tests.

## 3. Results

### 3.1. Results of Initial Analyses on Hukou Registration and Residence

Based on the initial analyses, the distribution of all respondents with different Hukou registration in rural and urban areas is shown in Table 1. In 2011, 24% of those with rural Hukou registration lived in urban areas, and 11% of those with urban Hukou registration lived in rural areas. In 2015, these proportions were 26% and 14%, respectively (Table 1). Hence, in view of our objective to identify rural residents (with rural Hukou registration, living in rural areas) and urban residents (with urban Hukou registration, living in urban areas), some 35 to 40% of respondents were misclassified, if only Hukou registration had been used.

Table 2 includes the results of the initial analyses of the prevalence rate of the health outcomes studied among respondents having concordant or discordant Hukou registration in rural and urban areas. The prevalence varied considerably between the different combinations of Hukou registration and actual residence. Respondents with rural Hukou registration who lived in urban areas had lower prevalence of the health problems than those living in rural areas.

Respondents with urban Hukou registration living in rural areas had higher prevalence of the health problems than those living in urban areas, except impaired IADLs in 2011. In the following results, we only focused on respondents with rural Hukou in rural areas (rural respondents) and respondents with urban Hukou in urban areas (urban respondents).

### 3.2. Prevalence of Health Outcomes in 2011 and 2015

Figure 2 shows the prevalence of all four health outcomes among rural and urban respondents, respectively, in 2011 and 2015. Rural respondents reported worse health outcomes than urban respondents. The prevalence of poor SRH and self-reported depression was statistically significantly lower in 2015 than in 2011, while the prevalence of impaired BADLs was significantly higher among both rural and urban respondents in 2015 than in 2011. There was no difference in the prevalence of impaired IADLs between the years among rural respondents or among urban respondents.

### 3.3. Individual Characteristics

In our study, the proportion of rural respondents was similar, nearly 73%, in 2011 and 2015. Rural respondents were slightly younger than urban respondents. There were more females among rural respondents than among urban respondents in 2015 (Table 3). Compared to urban respondents, more rural respondents were unmarried. In 2011, nearly 10% of rural respondents and nearly 14% of urban respondents lived alone (compared to 9% rural respondents and 8% urban respondents in 2015). Fewer respondents lived near their children in 2015 than in 2011 (Table 3).

Rural respondents had lower levels of education and income than urban respondents. In 2011, 67% of the rural respondents had less than primary school, compared to 27% of urban respondents. In 2015, this proportion was slightly lower, among both rural and urban respondents. The mean individual annual income among urban respondents was about four times higher than that among rural respondents (Table 3).

### 3.4. Prevalence of Poor SRH by Individual Characteristics

The results for poor SRH are presented in Table 4, Table 5 and Table 6. The results for the other outcomes are presented in the Supplement. Rural respondents reported higher prevalence of poor SRH than urban respondents (Table 4). The prevalence of poor SRH declined significantly from 39% in 2011 to 30% in 2015 among rural respondents and from 24% to 18% among urban respondents (Figure 2 and Table 4). Rural respondents had significantly higher prevalence of poor SRH than urban respondents within each age group (Table 4). Rural respondents aged 60–64 years had higher prevalence of poor SRH than urban respondents aged 85–89 years. The prevalence of poor SRH increased with age, except among respondents aged 90+ years. More females than males reported poor SRH, especially among rural respondents. Unmarried respondents had higher prevalence of poor SRH than married respondents. There was a gradient in poor SRH by education and income (Table 4).

### 3.5. Prevalence of Impaired BADLs by Individual Characteristics

The prevalence of impaired BADLs among the older respondents was slightly higher in 2015 than in 2011 (Appendix A). The prevalence of impaired BADLs increased with age. Compared to urban respondents, the impairments of BADLs appeared at younger age among rural respondents. The prevalence of impaired BADLs was higher among females and respondents living alone than among males and respondents not living alone. The prevalence of impaired BADLs was lower among groups with higher levels of education and income.

### 3.6. Prevalence of Impaired IADLs by Individual Characteristics

The prevalence of impaired IADLs increased with age (Appendix A). The impairments of IADLs among rural respondents appeared at younger age, compared to among urban respondents. There was no significant difference in the prevalence of impaired BADLs between rural and urban respondents who were aged 85 years and above in 2011 or in 2015. The prevalence of impaired IADLs was higher among females than among males. There was no significant difference between respondents living alone and not living alone. Groups with higher educational level and higher income level had lower prevalence of impaired IADLs. There was no significant difference in the prevalence of impaired IADLs between rural and urban respondents within each income group, except in the lowest income group in 2015 and in the highest income group in both years.

### 3.7. Prevalence of Self-Reported Depression by Individual Characteristics

Compared to urban respondents, nearly twice as many rural respondents reported depression (Appendix A). The prevalence rates of self-reported depression among both rural and urban respondents were significantly lower in 2015 than in 2011. Older respondents generally had higher prevalence of depression, but the prevalence was high also in the youngest age group of rural respondents. Respondents living alone, being unmarried, with lower educational and income level had higher rates of reporting depression. More females than males reported depression, especially among rural respondents.

### 3.8. Sociodemographic Determinants of Poor SRH, Self-Reported Functional Disabilities and Depression

Multiple logistic regression analyses were done to estimate to what extent differences in sociodemographic determinants could explain rural–urban differences in health outcomes. As shown in Table 5 (2011) and Table 6 (2015), in Model 1, after adjusting for age and sex, rural respondents had a higher OR of reporting poor SRH in 2011 (OR = 2.05) and in 2015 (OR = 2.00). In Model 2, further adjustment for marital status, living arrangement, living near children did not significantly change the OR for rural respondents. In Model 3, further adjustment for education level reduced the OR for having poor SRH among rural respondents in 2011 (OR = 1.66) and in 2015 (OR = 1.74). In Model 4, when instead adjusting for income group in 2011 the OR was lower than that in Model 3 (OR = 1.55). In the full model (Model 5), the OR for having poor SRH in 2011 (OR = 1.34) was further reduced compared to the previous model. In 2015, the proportion of respondents with missing information of income was more than 50%; hence income was not included in the regression analysis.

Similar findings were seen in the regression analyses for impaired BADLs, impaired IADLs, and self-reported depression (Appendix A, respectively). For impaired BADLs, in Model 1, rural respondents had higher OR of having impaired BADLs in 2011 (OR = 1.99) than in 2015 (OR = 1.76), Appendix A, respectively. In 2011, adjusting for living arrangement, living near children, educational level and income group, in Model 5, reduced the OR for having impaired BADLs to 1.09 (Appendix A). In 2015, the OR was reduced from 1.76 in Model 1 to 1.43 in Model 3. (Appendix A).

For impaired IADLs, in Model 1, the OR of rural respondents reporting impaired IADLs was 2.05 in 2011 and 2.03 in 2015, Appendix A, respectively. In 2011, adjusting for living arrangement, living near children, educational level and income group, in Model 5, reduced the OR for reporting impaired IADLs to 1.13 (Appendix A). In 2015, the OR after adjustment was 1.59 (Appendix A).

For self-reported depression, in Model 1, rural respondents had higher OR (2.36) in 2011 and in 2015 (2.32), Appendix A, respectively. In 2011, adjusting for living arrangement, living near children, educational level and income group, in Model 5, reduced the odds ratio for having self-reported depression to 1.38 (Appendix A). In 2015, the OR after adjustment was 1.84 (Appendix A).

## 4. Discussion

This study found substantially higher rates of adverse health outcomes among rural than urban respondents, both in 2011 and 2015. The prevalence rate of poor SRH, impaired IADLs, and self-reported depression was lower in 2015 than in 2011, but the prevalence rate of impaired BADLs was higher in 2015 than in 2011. However, the relative rural–urban differences in the health outcome measures were similar in both years. We used an alternative classification combining Hukou registration with actual residence to distinguish rural respondents from urban respondents. The usual classification, using only Hukou registration or actual residence, resulted in smaller rural–urban differences, i.e., an underestimation of rural–urban inequalities.

The main focus of our study was on rural–urban inequalities in self-reported health outcomes. Rural residents had twofold OR of poor SRH, impaired BADL, impaired IADL, and depression compared to urban residents. The rural–urban inequalities in health outcomes were to a large extent explained by rural–urban inequalities in educational level and income. This is in line with findings of a previous study which identified differences in socioeconomic status as a main pathway through which persons with rural Hukou and residence are disadvantaged in terms of health [10]. As in other studies [33,34], being married was associated with better health, but living alone or living near children was not significantly related to health outcomes.

Our results are otherwise in line with previous studies [5,21,24], and also consistent with lower life expectancy in rural compared to urban areas [35]. However, in China the social security policy has improved a lot. Both rural and urban residents are entitled to social security coverage such as health and pension insurance, but the insurance packages in rural and urban areas are still often different. Older people with rural Hukou living in urban areas can still access fewer social services because of their rural Hukou and lower income, even though they are entitled to the local social services.

Our health outcome measures included different aspects of functional abilities. Functional ability, the fundamental capability of individuals, is related to successful and positive aging [36]. Overall, difficulties in performing daily activities, both in BADLs and IADLs, were strongly associated with age, and these impairments appeared at a substantially younger age among rural respondents than among urban respondents. This suggests that the need of assistance because of functional limitations comes at an earlier age among rural residents. It also indicates the importance of prevention to reduce the occurrence of functional limitations.

The prevalence of impaired IADLs was higher than the prevalence of impaired BADLs among the rural respondents. Most studies assess functional abilities with BADLs [37,38,39], measured with basic activities to reflect primary biological and psychosocial function [40]. The IADLs measure acknowledges that complex functions normally are lost before BADLs [30], and are necessary for functioning in community settings, which may identify incipient decline in older adults [30]. However, few studies include IADLs in the concept of functional disability with BADLs [41,42,43]. Taking both into consideration may provide an overall assessment of the functional abilities among the older adults and could be useful when planning services for older adults in rural areas.

As in other studies, and similar to findings of a systematic review [44], rural older adults in our study had nearly two times higher prevalence of self-reported depression than urban older adults. The prevalence of self-reported depression was lower in 2015 than in 2011. Psychological health might be related to different factors, including improved health care access, social support and participation in the society [45].

The prevalence rates of all adverse health outcome measures, except impairments in BADL, declined from 2011 to 2015 among both rural and urban residents. The reasons for this improvement are likely manifold and a single cause cannot be pinpointed. However, from 2011 to 2015 there was an overall improvement in the Chinese economy [46]. The increase of income was also seen in our study. As all health outcome measures in our study were closely associated to the levels of education and income, improvements in education and income may have contributed to the improvements in health outcomes [47,48]. However, the relative rural–urban differentials in health were similar in 2011 and in 2015. Nevertheless, improvements in the level of education and income in rural areas may be an important part of policies to improve health among rural older adults and to reduce rural–urban inequalities in health, along with other improvements in access to health and social services.

### Strengths and Limitations of the Study

One strength of our study is that we used data from CHARLS, a nationwide representative sample derived from the large and diverse population in China and investigated different dimensions of self-reported health status, including general health, functional abilities, and psychological health. Furthermore, as one of the first studies, we used an alternative classification to investigate rural–urban inequalities, limiting the samples to respondents with rural Hukou living in rural areas and respondents with urban Hukou living in urban areas in order to control the influence of population mobility on rural–urban differences in health outcomes. Respondents with different Hukou have different entitlements regarding education, insurance, and health care. Among respondents with rural Hukou, almost 25% lived in urban areas, and more than 10% of respondents with urban Hukou lived in rural areas. In addition, there were significant differences in the health outcomes between respondents who lived in the same area but with different Hukou. This underlines the importance of taking not only Hukou registration but also actual residence into account, when comparing rural and urban residents. Using only Hukou registration would underestimate the differences between rural and urban respondents.

However, this study also has several limitations. The cross-sectional design of the study prevents causal inference. Also, we were not able to explore the effects of other sociocultural and environmental factors (e.g., access to clean water and good sanitation [49]) due to lack of specific data. Income was an important determinant of health and of rural–urban inequalities in health, especially in 2011. However, other studies have used household expenditure as a proxy for income, arguing that especially in rural areas many people do not have a monetary income but are reimbursed in kind for work in the agricultural sector [10,50,51]. Nevertheless, in our study income had a strong association to the health outcome measures investigated. Unfortunately, the importance of level of income could not be studied in 2015 because of the large proportion (nearly 60%) with missing data on income in 2015. Furthermore, this study attempted to examine rural–urban differences in health over time. The time span between the survey waves was only four years, which may be too short to show a trend. Nevertheless, the study might contribute to the sparse literature on whether rural–urban inequalities in China have changed over time.

Our study indicates that there are important differences in self-reported health, functional disabilities, and self-reported depression between rural and urban older adults, which to a considerable extent were explained by rural–urban inequalities in education and income. Further studies should investigate how health and social services might be strengthened in rural areas to improve health and functional abilities among rural older adults.

## 5. Conclusions

Using an alternative classification, combining Hukou registration and actual residence to distinguish rural and urban residents, we found large rural–urban inequalities in poor SRH, functional disabilities, and self-reported depression among older adults in China. These were to a great extent associated to rural–urban inequalities in educational level and income. Rural respondents had a higher prevalence of adverse health outcomes and the impairments appeared at a younger age among rural than among urban respondents. This study may serve as a basis for interventions to address rural–urban differences in health and social services and to reduce health inequalities among Chinese older adults.

## Figures and Tables

**Figure 1 ijerph-18-06557-f001:**
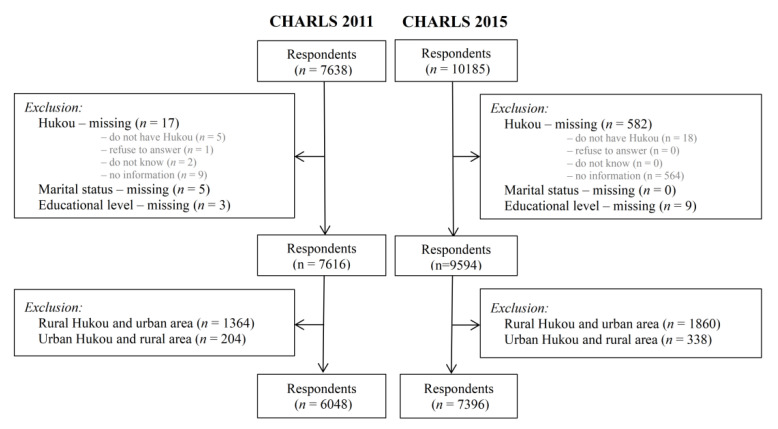
Flow diagram of samples, CHARLS (China Health and Retirement Longitudinal Study) 2011 and CHARLS 2015.

**Figure 2 ijerph-18-06557-f002:**
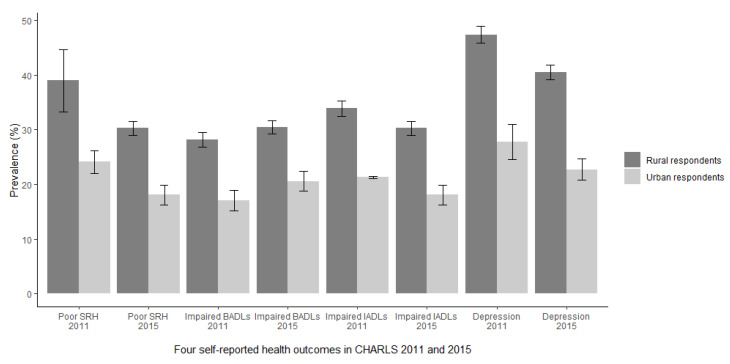
Prevalence (%) and 95% confidence interval (error bars) of poor Self-Rated Health (SRH), impaired Basic Activities of Daily Living (BADLs), impaired Instrumental Activities of Daily Living (IADLs), and self-reported depression among rural and urban respondents, CHARLS 2011 (*n* = 6048) and CHARLS 2015 (*n* = 7396).

**Table 1 ijerph-18-06557-t001:** Distribution (%) of respondents with different Hukou registration in rural and urban areas, CHARLS 2011 and CHARLS 2015.

Hukou Registration	Living Area	CHARLS 2011 (*n* = 7616)		CHARLS 2015 (*n* = 9594)	
		*n*	%	*n*	%
Rural Hukou	Rural area	4391	76.3	5385	74.3
	Urban area	1364	23.7	1860	25.7
Urban Hukou	Rural area	204	11.0	338	14.4
	Urban area	1657	89.0	2011	85.6

CHARLS: China Health and Retirement Longitudinal Study.

**Table 2 ijerph-18-06557-t002:** Prevalence (%) of health outcomes among respondents with different Hukou registration in rural and urban areas, CHARLS 2011 (*n* = 7616) and CHARLS 2015 (*n* = 9594).

	Living Area	Hukou Registration	Poor SRH ^a^		Impaired BADLs ^b^		Impaired IADLs ^c^		Depression	
			%	*p* *	%	*p* *	%	*p* *	%	*p* *
CHARLS 2011	Total		34.0		25.1		30.1		41.5	
	Rural area	Rural Hukou	39.0	<0.001	28.2	0.936	33.9	<0.001	47.4	<0.001
		Urban Hukou	26.1		27.7		18.8		27.0	
	Urban area	Rural Hukou	30.7	<0.001	24.4	<0.001	29.9	<0.001	41.3	<0.001
		Urban Hukou	24.1		17.1		21.3		27.8	
CHARLS 2015	Total		26.3		27.3		29.7		35.2	
	Rural area	Rural Hukou	30.3	<0.001	30.5	0.012	33.6	<0.001	40.5	0.005
		Urban Hukou	24.2		24.0		24.0		32.6	
	Urban area	Rural Hukou	23.6	<0.001	25.9	<0.001	28.9	<0.001	33.4	<0.001
		Urban Hukou	18.1		20.6		20.8		22.7	

^a^ SRH: Self-Reported Health; ^b^ BADLs: Basic Activities of Daily Living; ^c^ IADLs: Instrumental Activities of Daily Living; *** Chi-square test between respondents with different Hukou registration in rural and urban areas, respectively. The level of significance was specified at 0.05.

**Table 3 ijerph-18-06557-t003:** Characteristics of rural and urban respondents, CHARLS 2011 and CHARLS 2015.

	CHARLS 2011 (*n* = 6048)			CHARLS 2015 (*n* = 7396)		
	Rural (*n* = 4391)	Urban (n = 1657)	*p*	Rural (*n* = 5385)	Urban (*n* = 2011)	*p*
	%	%		%	%	
Age (mean ± SD ^a^, years)	68.3 ± 7.0	69.0 ± 7.0	0.001 *	68.6 ± 7.2	69.2 ± 7.2	0.001 *
Age group (years)			0.009 **			0.001 **
60–64	38.3	34.3		35.8	32.7	
65–69	24.6	23.2		26.9	26.3	
70–74	17.3	20.0		17.7	17.3	
75–79	11.4	13.7		10.5	14.2	
80–84	5.7	6.3		5.9	6.3	
85–89	2.0	2.1		2.5	2.5	
90+	0.6	0.5		0.8	0.7	
Sex			0.111 ***			0.006 ***
Male	49.6	51.9		48.3	51.9	
Female	50.4	48.1		51.7	48.1	
Marital status			<0.001 ***			<0.001 ***
Married	76.7	81.1		77.3	81.6	
Unmarried	23.3	18.9		22.7	18.4	
Living arrangement			<0.001 ***			0.234 ***
Not living alone	90.5	86.3		91.1	91.9	
Living alone	9.5	13.7		8.9	8.1	
Living near children			<0.001 **			0.818 ***
Living near children	90.9	91.9		87.4	87.9	
Not living near children	6.9	4.7		10.5	10.2	
No child	2.2	3.4		2.0	1.8	
Educational level			<0.001 **			<0.001 **
Below primary school	67.2	26.6		64.9	22.0	
Primary school	23.5	23.9		22.8	27.5	
Middle school	7.8	24.6		9.8	26.0	
High school	1.5	16.5		2.4	17.3	
College and above	0.1	8.3		0.1	7.2	
Income (mean, CNY ^b^)	4126	18,846	<0.001 *	5294	20,979	<0.001 *
Income group			<0.001 **			<0.001 **
First group (low)	21.0	4.0		10.4	5.3	
Second group	22.8	2.2		11.8	1.3	
Third group	20.2	6.1		11.9	1.9	
Fourth group	13.9	18.9		10.0	7.0	
Fifth group (high)	6.2	50.2		4.1	24.4	
Missing	15.9	18.7		51.8	60.1	

^a^ SD: Standard Deviation; ^b^ CNY: Chinese Yuan; * Independent *t* test; ** Chi-square test. The level of significance was specified at 0.05. A Bonferroni adjusted significance level was used in case of multiple tests; *** Chi-square test.

**Table 4 ijerph-18-06557-t004:** Prevalence (%) of poor Self-Rated Health (SRH) among rural and urban respondents, CHARLS 2011 and CHARLS 2015, by individual characteristics.

	CHARLS 2011 (*n* = 6003)			CHARLS 2015 (*n* = 6844)		
	Rural (*n* = 4370)	Urban (*n* = 1633)	*p* *	Rural (*n* = 5018)	Urban (*n* = 1826)	*p* *
Total	39.0	24.1	<0.001	30.3	18.1	<0.001
Age group (years)						
60–64	35.5	20.0	<0.001	27.1	16.8	<0.001
65–69	35.9	26.6	0.001	30.7	14.8	<0.001
70–74	42.0	26.1	<0.001	32.2	22.4	0.001
75–79	45.3	24.4	<0.001	36.9	21.2	<0.001
80–84	48.0	27.5	0.001	29.7	17.3	0.016
85–89	57.5	30.3	0.013	38.8	28.1	0.282
90+	40.7	37.5	1.000	26.7	16.7	1.000
Sex						
Male	35.5	21.4	<0.001	27.9	15.9	<0.001
Female	42.4	27.1	<0.001	32.6	20.5	<0.001
Marital status						
Married	38.1	24.3	<0.001	28.7	17.5	<0.001
Unmarried	42.0	23.3	<0.001	36.4	21.2	<0.001
Living arrangement						
Not living alone	39.1	24.3	<0.001	30.3	18.0	<0.001
Living alone	38.2	22.7	<0.001	30.2	20.3	0.024
Living near children						
Living near children	39.0	24.0	<0.001	30.6	18.2	<0.001
Not living near children	39.2	23.1	0.008	26.8	16.9	0.007
No child	38.7	29.4	0.265	38.3	25.0	0.225
Educational level						
Below primary school	41.9	31.4	<0.001	32.2	23.8	0.001
Primary school	35.0	26.9	0.004	28.0	18.1	<0.001
Middle school	29.1	20.3	0.006	26.5	16.0	<0.001
High school	29.2	18.1	0.045	19.8	16.6	0.419
College and above	0.0	16.2	1.000	0.0	13.1	0.344
Income group						
First group (low)	45.4	39.4	0.344	33.2	21.7	0.020
Second group	41.6	36.1	0.524	31.3	26.9	0.638
Third group	35.9	33.7	0.652	32.2	34.2	0.801
Fourth group	32.8	25.9	0.029	28.0	25.5	0.561
Fifth group (high)	27.7	20.9	0.020	14.8	12.8	0.475
Missing	40.7	23.3	<0.001	30.8	18.5	<0.001

***** Chi-square test. The level of significance was specified at 0.05.

**Table 5 ijerph-18-06557-t005:** Multiple logistic regression analyses on rural and urban respondents’ poor Self-Rated Health (SRH), adjusted for age and sex, CHARLS 2011 (*n* = 6003).

	Model 1		Model 2		Model 3		Model 4		Model 5	
	OR	95% CI	OR	95% CI	OR	95% CI	OR	95% CI	OR	95% CI
Rural respondents ^a^	2.05	1.80–2.33	2.04	1.79–2.32	1.66	1.43–1.92	1.55	1.32–1.81	1.34	1.13–1.58
Marital status ^b^										
Unmarried	—	—	0.99	0.84–1.16	0.97	0.83–1.14	0.99	0.84–1.17	0.98	0.83–1.15
Living arrangement ^c^										
Living alone	—	—	0.81	0.65–1.00	0.80	0.65–0.99	0.74	0.59–0.92	0.74	0.59–0.92
Living near children ^d^										
Not living near children	—	—	1.10	0.88–1.37	1.10	0.89–1.38	1.07	0.85–1.33	1.07	0.86–1.34
No child	—	—	1.25	0.87–1.79	1.23	0.86–1.76	1.26	0.88–1.81	1.29	0.87–1.79
Educational level ^e^										
Primary school	—	—	—	—	0.85	0.74–0.98	—	—	0.86	0.75–0.99
Middle school	—	—	—	—	0.66	0.54–0.80	—	—	0.69	0.56–0.84
High school	—	—	—	—	0.57	0.43–0.77	—	—	0.63	0.46–0.85
College and above	—	—	—	—	0.45	0.28–0.72	—	—	0.49	0.30–0.80
Income group ^f^										
Second group	—	—	—	—	—	—	0.87	0.73–1.04	0.87	0.73–1.04
Third group	—	—	—	—	—	—	0.71	0.59–0.85	0.71	0.59–0.85
Fourth group	—	—	—	—	—	—	0.62	0.52–0.76	0.63	0.52–0.76
Fifth group (high)	—	—	—	—	—	—	0.48	0.39–0.60	0.52	0.42–0.65
Missing	—	—	—	—	—	—	0.76	0.63–0.92	0.78	0.64–0.94

OR: odds ratio; CI: confidence interval 95%; ^a^ Reference group: urban respondents; ^b^ Reference group = unmarried; ^c^ Reference group = not living alone; ^d^ Reference group = living near children; ^e^ Reference group = below primary school; ^f^ Reference group = first group (low).

**Table 6 ijerph-18-06557-t006:** Multiple logistic regression analyses on rural and urban respondents’ poor Self-Rated Health (SRH), adjusted for age and sex, CHARLS 2015 (*n* = 6844).

	Model 1		Model 2		Model 3	
	OR	95% CI	OR	95% CI	OR	95% CI
Rural respondents ^a^	2.00	1.75–2.28	1.98	1.73–2.26	1.74	1.49–2.02
Marital status ^b^						
Unmarried	—	—	1.42	1.21–1.67	1.40	1.19–1.65
Living arrangement ^c^						
Living alone	—	—	0.68	0.54–0.85	0.68	0.53–0.85
Living near children ^d^						
Not living near children	—	—	0.94	0.79–1.13	0.95	0.79–1.14
No child	—	—	1.56	1.05–2.31	1.52	1.02–2.26
Educational level ^e^						
Primary school	—	—	—	—	0.85	0.74–0.98
Middle school	—	—	—	—	0.81	0.67–0.97
High school	—	—	—	—	0.71	0.54–0.94
College and above	—	—	—	—	0.53	0.31–0.91

OR: odds ratio; CI: confidence interval 95%; ^a^ Reference group: urban respondents; ^b^ Reference group = unmarried; ^c^ Reference group = not living alone; ^d^ Reference group = living near children; ^e^ Reference group = below primary school.

## Data Availability

The CHARLS data set is publicly available. Information about the data source and available data is found at http://charls.pku.edu.cn/pages/data/111/en.html, accessed on 3 November 2020. Researchers can obtain these data after submitting a data use agreement to the CHARLS team.

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
