# Peer review of "Rural–Urban Inequalities in Poor Self-Rated Health, Self-Reported Functional Disabilities, and Depression among Chinese Older Adults: Evidence from the China Health and Retirement Longitudinal Study 2011 and 2015"

_ijerph, 2021, doi:10.3390/ijerph18126557_

Round 1
Reviewer 1 Report
This is an interesting paper by Jiang et al regarding the rural-urban inequalities in self-rated health, functional abilities, and depression among older Chinese adults. The study was based on the China Health and Retirement Longitudinal Study and data from 2011 and 2015 were compared. The authors report that there are inherent inequalities in the provided entitlements between the Hokou system, with people in rural areas having less care, although they seem to have worse health indices. Another problem identified is that people from rural areas migrating to urban areas, cannot change their status and carry their rural health coverage. This seems to affect their health status in the urban area they chose to live in, suggesting that this could lead to major inequalities between citizens, solely due to their initial registration in the Huckou systemrural descent. The authors base their conclusions on the study of the actual residence of the test subjects and not the Hackou reported rural/urban status.
The study provides a novel, although limited in size, insight into the effect of income inequalities in China on these health indices. It verifies what is known in western societies that poverty should be considered an independent factor for poor health outcomes and it is very interesting to also evaluate this in a country like China from a rural/urban perspective.
The major issue regarding this study is that the authors, after an initial presentation of the problem generated by rural to urban (and vice versa) migration in table 2, they solely focus on people with rural residence and rural Huckou vs people with urban residence and urban Huckou. Although this is clearly interesting, it is arbitrary and only describes part of the problem. The issue of the inequality of health coverage between citizens that live in the same area but have different original Hockou registration and its potential effect on the studied indices is included the data, is of great importance and should be presented. If this is not possible, a clear explanation should be provided for not performing this analysis.
Minor issues
- Please clarify the Bonferroni adjusted significance level in table 3 and 4
- Table 2 and Figure 2 seem to include the same data and it is unclear what the error bars represent, or how they were calculated.
Reviewer 2 Report
Well presented analysis on older/dated survey results. Certainly the study has several limitations, but the authors did a good job of including those and not overstating their conclusions. The value of the study is limited, but the manuscript is solid.
Round 2
Reviewer 1 Report
The authors have responded properly to my previous remarks.
Minor point:
Please provide acronyms' explanations once. Especially in Tables 2a and 2b, explaining acronyms twice feels unnecessary.
